# Subchondral Bone Cyst Development in Osteoarthritis: From Pathophysiology to Bone Microarchitecture Changes and Clinical Implementations

**DOI:** 10.3390/jcm12030815

**Published:** 2023-01-19

**Authors:** Angelos Kaspiris, Argyris C. Hadjimichael, Ioanna Lianou, Ilias D. Iliopoulos, Dimitrios Ntourantonis, Dimitra Melissaridou, Olga D. Savvidou, Evangelia Papadimitriou, Efstathios Chronopoulos

**Affiliations:** 1Laboratory of Molecular Pharmacology, Department of Pharmacy, School of Health Sciences, University of Patras, 26504 Patras, Greece; 2Department of Orthopaedics, St Mary’s Hospital, Imperial College Healthcare NHS Trust, Praed Street, London W2 1NY, UK; 3Department of Orthopaedic Surgery, “Rion” University Hospital and Medical School, School of Health Sciences, University of Patras, 26504 Patras, Greece; 4Accident and Emergency Department, “Rion” University Hospital and Medical School, School of Health Sciences, University of Patras, 26500 Patras, Greece; 5First Department of Orthopaedic Surgery, School of Medicine, National and Kapodistrian University of Athens, “ATTIKON” University General Hospital, Rimini 1, 12462 Athens, Greece; 6Laboratory for Research of the Musculoskeletal System, School of Medicine, National and Kapodistrian University of Athens, 14561 Athens, Greece

**Keywords:** subchondral cyst, cartilage, osteoarthritis, bone turnover, angiogenesis

## Abstract

Osteoarthritis is a degenerative joint disease affecting middle-aged and elderly patients. It mainly involves weight-bearing joints such as the hip, knee and spine as well as the basilar joint of the thumb, causing dysfunction and painful symptoms. Often, joint arthritis is accompanied by cartilage defects, joint space narrowing, osteophytes, bone sclerosis and subchondral bone cysts (SBC). The aim of the present study was to explore the pathophysiology responsible for the development of SBCs as well as the association between SBCs and disease progress, the level of clinical symptoms and their impact on postoperative outcomes and risk of possible complications following joint replacements if left untreated. A literature review on PubMed articles was conducted to retrieve and evaluate all available evidence related to the main objective mentioned above. A few theories have been put forth to explain the formation process of SBCs. These involve MMPs secretion, angiogenesis, and enhanced bone turnover as a biological response to abnormal mechanical loads causing repeated injuries on cartilage and subchondral tissue during the development of arthritis. However, the application of novel therapeutics, celecoxib-coated microspheres, local administration of IGF-1 and activated chondrocytes following surgical debridement of SBCs hinders the expansion of SBCs and prevents the progression of osteoarthritis.

## 1. Introduction

Osteoarthritis (OA) is one of the leading causes of disability in the adult population. Although it was previously considered primarily as an articular cartilage disorder, it is now recognised as a type of panarthritis—a condition involving all the structures of a joint, including the calcified cartilage, the subchondral bone, the capsular ligaments and the synovial fluid [1].

Newer data show that the integrity and functionality of the articular cartilage can be significantly affected by pathological conditions that cause damage to other joint tissues [2,3]. In addition to its very important structural support role, the subchondral bone presents an equally important biological crosstalk with the overlying cartilage, making the articular cartilage–subchondral bone system a single inseparable functional unit [4]. A characteristic change of the subchondral bone in the context of OA is the creation of subchondral bone cysts (SBCs) [5]. SBCs occur frequently in degenerative joint diseases and are closely related to OA, while bibliographic data indicate that they are found in 50% of patients with knee joint OA [6,7]. In most cases, these are spherical or ellipsoidal gelatinous sacs ranging from a few millimetres to 5 centimetres in diameter. They are detectable by a standard radiograph in the subchondral bone of the articular surfaces of the large joints, and they develop due to the bone’s attempt to adapt in areas subject to increased loads [8,9].

Based on the histological characteristics of SBCs, it could be noted that the term ‘cyst’ is not entirely accurate, since these cavernous lesions are not covered by epithelial cells and do not contain uniformly distributed liquid elements [10]. A layer of fibrous connective tissue covered by lining cells and osteoblasts may in some cases surround myxoid or adipose tissue and cartilage; this content may at first be liquid and then become ossified [11].

A histological study of human bone preparations from osteoarthritic knees [12] showed that SBCs may also contain fragments of necrotic bone tissue and dead apoptotic cells, i.e., cells without nuclei. In addition, immunohistochemical techniques in animal preparations [13] have highlighted a high number of osteoblasts and osteoclasts, which is considered indicative of the high rate of bone reconstruction and mineralisation characteristic of SBCs.

SBCs are imaged as well-delimited cavities of fluid signal on magnetic resonance imaging (MRI), corresponding to radiolucency encompassed with sclerotic margins on standard radiographs [12]. These are usually either multiple microscopic lesions with diameters of 2 to 15 mm or single lesions of up to 2 cm and more rarely larger, appearing as cystic spaces between thinning bone trabeculae [14].

The radiological characteristics of SBCs distinguish them from local tumours, although differential diagnosis may be confusing in certain cases [15]. Giant cell tumours occur between the ages of 20 and 30 years, are solitary, eccentric, without sclerotic margins and are mainly located in bone metaphyses, extending to the epiphysis of the bone. Chondroblastoma, which is also a type of solitary tumour, is found in younger people (10 to 30 years), while clear-cell chondrosarcomas have a cartilaginous background with calcification elements. Finally, metastatic bone lesions can be solitary or multiple and occur in older patients, and they do not present degenerative elements [16].

MRI is the imaging method of choice for the evaluation of cartilage damage and the investigation of bone marrow lesions [8]. Specifically, MRI shows increased sensitivity for the detection of very small SBCs (well-delimited rounded lesions with fluid-like signal intensity in sequences without a contrast agent) in early-stage OA, much before the appearance of arthritic joint findings on X-rays [8]. Data from the MOST study of Crema et al. [8] showed that in several cases, signal enhancement was observed after the administration of an intravenous paramagnetic agent. Carrino et al. [17] used MRIs to study the relationship between SBCs and areas of bone marrow oedema. Examining data from 32 patients with knee OA, they concluded that SBCs are formed in areas displaying a signal indicating bone oedema in the MRI, while the overlying articular cartilage shows significant deterioration. According to this longitudinal assessment of knee MR images, the presence of bone marrow oedema areas that are subjacent to cartilage abnormalities can be characterised as pre-SBCs changes in OA patients [17]. Therefore, the early detection of bone marrow “oedema-like” signals on MRI is able to predict the formation of SBCs, causing pain and discomfort in arthritic knees, in the future [17].

The use of CT scan highlights the key points of advanced OA, including osteophytes, SBCs and subchondral bone sclerosis, and it can also contribute to the understanding of OA pathophysiology by measuring the density and structure of subchondral mineralised tissues in vivo [18].

In recent years, these sacs have attracted the interest of the scientific community due to the possible association of their presence with increased articular cartilage loss and an increased risk for joint replacement surgery [19]. The study aims to examine the association of SBC development with altered angiogenesis and bone turnover as well as their clinical impact during OA progression.

## 2. Materials and Methods

### 2.1. Research Strategy

The authors investigated published studies reporting the development of SBCs in osteoarthritis. A specific interest of the present study was to identify the pathophysiologic process that contributes to bone microarchitecture alterations and the clinical impact of SBCs. A systematic computer-based literature review search with predefined criteria was performed in MEDLINE/PubMed (1946 to present) of the National Library of Medicine and EMBASE (1947 to present). The combination of the following terms was used: “subchondral cyst (All fields)”, “cartilage (All fields)”, “osteoarthritis (All fields)”, “bone turnover (All fields)”, “angiogenesis (All fields)”, “diagnosis (All fields)” “prognosis (All fields)”, and “treatment (All fields)”. The electronic literature was searched independently by two authors (A.K and A.C.H.). The tables and images were designed by author A.K. The final article was reviewed and approved by the senior supervisors (O.D.S., E.P., E.C.).

### 2.2. Inclusion and Exclusion Criteria

Published articles written in English in peer-reviewed journals were considered. These articles were either literature reviews or clinical studies on the pathophysiology related to the development of SBCs in osteoarthritis and their clinical implementation. Case reports and preclinical studies describing the application of novel techniques for the treatment of SBCs were also included. In addition, the articles and abstracts were screened by the two senior authors (A.K., and A.C.H.) independently. All identified studies were evaluated by the authors (I.D.I., D.N., I.L. and D.M.), and data were extracted using a predetermined form.

Articles written in a language other than English, expert opinion reports, letters to the editor, posters, presentations, duplicate studies and surveys with insufficient data on the pathophysiology and clinical manifestations of SBCs in arthritis were excluded. Articles exhibiting the impact of cystic lesions except for common SBCs, such as benign lesions (chondroblastoma, giant cell tumour), were also excluded. The Endnote software (Clarivate Analytics, Philadelphia, PA, USA) was used to assess the presence of duplicate studies, which were eventually excluded (Figure 1).

## 3. Results

### 3.1. Pathophysiology

SBCs are referred to in the literature under the following terms: pseudocysts, osteoarthritic cysts, geodes, or Egger cysts (in the hip socket). They were first identified by Ondrouch [20] and Landells [21] in the load-bearing regions of the femur, patella, and shoulder joints of arthritic patients. Although their pathogenic mechanism has not yet been elucidated, there are two prevalent theories regarding the formation of SBCs in load-bearing bone regions in the context of OA [19,20,22].

According to the first theory (Synovial Breach Theory) [11], SBCs are created due to the loss-thinning of the supernatant articular cartilage and repeated microtrauma to the osteochondral junction. In this case, microfractures in the articular surface allow the penetration of synovial fluid and/or tissue into the area of the subchondral bone, leading to an acute inflammatory response and the development of myxomatous tissue within the bone marrow.

In contrast, according to the second theory (Bone Contusion Theory) [23], there is no direct communication between the joint and the subchondral bone. The wear and subsequent weakening of the articular cartilage leads to uneven distribution of loads and degradation of the quality of the underlying bone, resulting in microfractures of the subchondral bone and bone marrow oedema, activating the bone reconstruction process. The aggregation of osteoclasts and macrophages causes bone absorption and phagocytosis of necrotic elements, with subsequent recruitment of osteoblasts for the deposition of new bone, which eventually encapsulates a cavity with fibrous content, creating the SBC.

However, in some cases, the creation of SBCs cannot be explained by any of the above pathogenic mechanisms. Pouders et al. [12] studied 42 bone preparations of tibial condyles from patients undergoing total knee arthroplasty due to severe OA. They observed a typical occurrence of SBCs in the intercondylar fossa, which is a part of the bone that is not covered by articular cartilage and is not a load-bearing area of the knee, in 45% of cases. This fact negates both the theory of synovial fluid leakage through microfractures in the cartilage as well as the theory of bone damage due to unequal load distribution. The authors speculated that SBCs in the intercondylar fossa are likely to be the result of the repetitive force exerted by the cruciate ligaments attached to the area.

In a 2017 study, Chan et al. [24] tried to explain this phenomenon by adding a third hypothesis, which assumes that the pathogenetic mechanism involves vascular factors. They formulated the theory that on non-loaded surfaces, SBCs are primarily the result of ischaemia of the subchondral bone due to inadequate irrigation in the context of ageing and dysfunction of the endothelium. By using immunohistochemical techniques, they presented their observations on the possible association between vascular pathology and osteoarthritic lesions, including SBCs. This resulted in a proposal to use the radiological findings of an SBC identified in a non-loaded area to identify a subclinical underlying vascular disorder. According to Shabestari et al. [25], angiogenesis was significantly enhanced in bone marrow lesions, which are considered the precursor stage of SBC formation at the base of osteoarthritic thumbs and hips. By using the dynamic histomorphometry technique, they compared the bone turnover and angiogenesis in the excised trapeziums (thumb OA) and the replaced femoral heads (hip OA) in patients with identified bone marrow lesions, with their counterparts with hip OA and no lesions. Based on their observations, the increased vascularity of SBCs supports the concept that subchondral bone is a highly mechanoresponsive tissue that responds to the biochemical changes of the chronic degeneration of the covering cartilage by increasing angiogenesis and attributing it to the formation of SBCs [24].

Matrix metalloproteinases (MMPs) are a family of Zn^++^-dependent proteases that degrade the extracellular matrix and enhance the angiogenic process by facilitating the invasion of endothelial cells. Kaspiris et al. [16] examined the association of the presence of SBCs with the expression of interstitial MMP-1 in patients with idiopathic hip or knee OA. The research showed that SBCs are associated with advanced OA and contain activated cells expressing MMP-1, with a possible active involvement in the osteochondral changes that occur during the progression of OA. The increased expression of MMPs by osteoblasts and by bone lining cells or silent SBC wall osteoblasts was also confirmed in further research by the same research group, concerning MMP-12 (Figure 2) [26].

### 3.2. Subchondral Bone Cysts and Bone Turnover

Two distinct phases, a destructive and a productive, can be observed in the SBC formation process [11]. The destructive phase begins with the osteonecrosis of the subchondral trabecular bone and ends with the subsequent bone absorption and the creation of a cavity with fibrous content (Figure 3). The productive phase concerns the process of the formation of sclerotic tissue at the borders of the fibrous cavity, with a simultaneous increase in both the density and volume of the cyst (Figure 4). In contrast to the increased density, a corresponding decrease in mineralisation is observed, which can lead to degraded stability in the area and a compensatory overall increase in density and bone sclerosis [27,28,29,30,31,32,33] (Table 1).

Chen et al. [34] studied the process of bone reconstruction in the context of articular cartilage destruction in patients with knee OA (Figure 3). Of the 97 patients in the study, 74.2% presented SBCs with findings of a sclerotic bone sheath of the cysts with increased bone volume and trabecular stiffness. Histological examination revealed the presence of cartilaginous tissue both in the perimeter and within the SBC, which consisted of type I collagen fibres. Type I fibres appear characteristically in the fibrous cartilage as opposed to the type II collagen fibres found in the hyaline cartilage of joints, which is a finding that reinforces the theory of bone damage in the context of the pathogenic mechanism of SBCs (Figure 2).

Indeed, ex vivo studies both in the hip joint using high-resolution quantitative CT (HR-qCT) [13] and in the tibia using micro-computational CT (micro-CT) [35] report changes in bone mineral density (BMD), especially in areas around SBCs. However, even though these methods are very accurate in distinguishing the quantitative and qualitative characteristics of both SBCs and the local subchondral bone, they cannot correlate these findings with critical clinical manifestations such as pain, severity and disease progression.

To answer this question, in a recent research study, Burnett et al. [36] used the qCT method to examine the characteristics of proximal tibia SBCs in patients with knee OA in vivo and to investigate their possible association with the clinical manifestations of the condition. The authors reported a wide variation in the number and volume of SBCs in the patient sample tested, suggesting that findings can be highly diverse from patient to patient. However, according to the results of the study, the largest number and volume of SBCs were associated with a higher BMD in both the medial and lateral compartments of the proximal tibia, while SBCs of the outer compartment in particular were associated with specific characteristics such as gender, the mechanical axis of the joint and the severity of OA changes in the outer compartment. However, they were not able to correlate any characteristic of SBCs with the occurrence of pain in patients with advanced OA.

The first study in knee OA patients, which correlated serum bone turnover biomarkers with the presence of painful symptoms and MRI findings, one of them being SBCs, was conducted by Hick et al. [28]. Specifically, the serum cartilage biomarkers Coll2-1 and Coll2-1NO2 were evaluated using immunoassay techniques, and they were found to have significantly increased in OA knee patients who had MRI findings such as subarticular bone attrition, periarticular cysts/bursitis and SBCs. In addition, these biomarkers were associated with the level of pain and dysfunction in patients suffering from knee OA [28].

Moreover, the formation of SBCs depends on the local bone turnover and the hypothesis of synovial intrusion [27]. According to that hypothesis, the synovial fluid fills a trabecular compartment that appears because of cartilage loss and subchondral bone attrition. The increasing mechanical loads on the trabeculae bone, as well as the decreased perfusion and oxygen supply, cause bone remodelling, necrosis of osteocytes and the formation of the cavity-like microarchitecture of SBCs [27]. According to Nakasone et al. [27], SBCs have heterogeneous characteristics either with fibrous (most common) or fatty content. Based on their histological and micro-computed tomography findings, SBCs usually contain neo-angiogenic vasculature and nerve fibres, cartilage islands and bony spicules. Furthermore, the local intense bone turnover makes the bone adjacent to SBCs denser (increased mineralisation) and cartilage overlying SBCs more thickened compared to non-overlying regions, which is in line with the findings by Shabestari et al. [25,27].

An experimental study by Von Rechenberg et al. [37] elucidated some of the cellular mechanisms responsible for the formation and pathogenesis of SBCs in OA patients. According to their findings, the fibrous tissue of SBCs produces oxide (NO), prostaglandin E2 (PGE2), and MMPs, such as collagenases and gelatinases (Figure 2), which promote pathological bone absorption by activating osteoclasts [37]. Interestingly, the concentration of NO, PGE2 and MMPs was found significantly increased in SBCs compared to normal joints and joints with severe OA, suggesting that moderate joint OA is biochemically more active compared to normal joints and those with severe OA (Figure 2). These secreted mediators maintain and decrease the healing rate of SBCs. In addition, when SBCs communicate with the arthritic joint, the release of bone-absorbing molecules from cartilage and synovial membranes increases bone absorption, leading to the progression of early-stage to end-stage OA [33]. Another study, by Sabokbar et al. [38], showed that in osteoarthritic hip acetabula, the local increase in mechanical pressure evokes the high infiltration of macrophages within SBCs. The cellular alteration of macrophage–osteoclast differentiation may enhance the OA subchondral bone remodelling and osteoclastic resorption of the subchondral bone surrounding the OA SBCs [38]. The macrophage colony-stimulating factor (M-CSF) is endogenously secreted by the macrophage progenitors in SBCs, inducing their differentiation into mature osteoclasts and causing the development of the cystic areas within the continuously absorbed osseous tissue [38].

As already mentioned, the formation of SBCs is associated with increased loads in the affected area, which can activate the bone reconstruction process. Additionally, it is well established that the OA of a joint induces the development of SBCs and vice versa. According to McErlain et al. [5], SBCs accelerate the progress of early-stage knee OA by increasing the peri-cystic intra-osseous stress up to 81% and causing greater pain and disability in patients. To the best of our knowledge, the extensive loading and recurrent microfractures in the subchondral bone due to a dysregulation of the mechanical and anatomic axes in OA joints (i.e., varus or valgus deformity in knee joints) allows the flow of synovial fluid beneath the cartilage and the formation of SBCs [5]. Furthermore, the expansion of SBCs and the formation of sclerotic bone (Figure 4) along their surface is one of the mechanisms that help SBCs to expand, increase the damage of subchondral bone, and enhance the calcification of cartilage, deteriorating its elastic properties and accelerating the development of OA to more severe stages [5]. The cause of SBC formation in young populations is due to mechanical factors (sports injuries); however, in adults and elderly patients suffering from OA, the altered mechanical forces arise within their joints [39]. Frazer et al. [39] investigated the role of tensile stresses and strains, compressive stresses and strains, and shear and von Mises stresses in the development and expansion of SBCs [39]. Based on their experimental evidence, the local stress-induced damage of the subchondral bone initiates the formation of SBCs. Furthermore, the presence of SBCs in the trabeculae of the subchondral bone generates additional mechanical stress risers, leading to the enlargement and increasing the diameter of SBCs [39].

### 3.3. Clinical Relevance

Pathological imaging findings, such as SBCs (Figure 4), are very commonly found in OA in combination with articular cartilage thinning, subchondral bone thickening, osteophyte formation, varying degrees of joint inflammation, ligament and meniscal degeneration and synovial membrane hypertrophy. However, how much does the presence of SBCs affect the clinical presentation of the patient? The modern definition of OA should include both patient-reported symptoms and structural changes within the joint [2].

There are many studies [40,41,42] examining the parameters associated with the progression of OA and the relationship between structural characteristics and pain. The results are mixed regarding the correlation of both the severity of the pain, especially in the knee joint, and the progression of the disease concerning the radiological findings and especially the presence of SBCs. Meniscal degeneration, full-thickness cartilage loss and osteophytes appear to be more associated with a more painful clinical image of the knee compared to the presence of SBCs in patients with OA [19,42,43].

However, the presence of SBCs is associated with the degradation of the articular cartilage, both quantitatively and qualitatively, which increases the risk that these patients will soon have to undergo total arthroplasty. The existence of SBCs may not be directly related to the clinical presentation, but it may identify patients with the worst structural deficits, especially in the knee joint, including increased cartilage loss, compared to patients who have only BMLs and who may benefit more from preventing disease progression [19].

Regardless of their aetiology and the role they play in the clinical presentation of a patient with OA, SBCs tend to appear in the advanced stages of the disease, where the degree of joint destruction combined with the poor clinical presentation leads the patient to the solution of total arthroplasty [44]. However, can the presence of SBCs affect the outcome of joint replacement surgery both directly and in the long term? According to the literature, smaller cysts usually regress after total hip arthroplasty or remain at their original size without affecting the functional outcome and viability of the prostheses. The larger ones usually do not regress and, if not treated intraoperatively, do not seem to affect the outcome of the surgical intervention [30]. Similar results are described by Takada et al. [31], who report that SBCs in the acetabulum of patients with congenital hip dysplasia undergoing total arthroplasty do not appear to increase in size nor affect the incorporation of prostheses and the long-term survival of the operation. Furthermore, a retrospective study by Kelly et al. [33] examined the impact and natural history of SBCs during 130 uncemented THA. According to their observations, no statistically significant difference was found among SBCs regarding initial size, patient age, patient weight, different cup types, gender and application of bone grafting after a 10-year follow-up [33]. However, the authors proposed the placement of the cup to be sealed and cover the SBCs alongside grafting at the time of primary THA to prevent possible progression and osteolysis. To our knowledge, the connection between the joint and SBCs may increase the influx of fluid to the cysts, causing their expansion and development to larger ones, thus increasing the risk of osteolysis and implant loosening [33].

A regression of SBCs has also been observed in the knee joint of patients who have undergone high tibial osteotomy, which, according to Wang et al. [29] appears to be due to the restoration of normal loads on the subchondral bone after this intervention. The authors conclude that one of the most likely explanations for this phenomenon could be the application of Wolff’s law.

Likewise, a prospective study by Mechlenburg et al. [32] revealed that cartilage thickness, as well as SBC volume, were not affected after a 10-year follow-up in patients who had periacetabular osteotomy (PAO) as an early treatment of hip dysplasia. As reported by the authors, these patients with unchanged SBCs did not experience substantial hip pain, and THA was not required [32]. Subsequently, a long-term follow-up in patients with a history of PAO revealed excellent results significantly correlated with the halt of progression that is often observed in OA if the disease remains untreated, and therefore, SBCs exhibit ongoing expansion in association with pain and reduced joint range of motion.

Preoperative surgical planning for the treatment of OA is crucial, and the outcomes may be affected by the presence of concurrent SBCs. Tanner et al. [45] examined the impact of SBCs on the risk of failure and loosening of implants after total shoulder arthroplasty (TSA) in OA patients. It is well known that the long-term survival of the glenoid implant in TSA relies on its fixation, which includes cement fixation and bony ingrowth to achieve sufficient stability during excessive shoulder movements [45]. According to the findings by Tanner et al. [45], the risk for revision was higher in medium to large (median volume: 518.7 to 1215.5 mm^3^) glenoid SBCs compared to the absence of or small (median volume: 130.4 mm^3^) SBCs which were identified with a quantitative CT scan preoperatively. However, the presence or not as well as the different volumes of SBCs were not correlated with differences in functional outcomes and component loosening rates in the implants of these patients [45]. Subsequently, the higher revision rates in the existence of medium to large SBCs were associated with the ability to achieve stable fixation of the glenoid component by the shoulder surgeons. Care must be taken in the preoperative planning of joint arthroplasties, and special imaging must be carried out to determine if implants are very close to SBCs, as the revision rate may be extremely increased. A case report by DiDomenico et al. [46] described the need for revision surgery after a total ankle arthroplasty in a 36-year-old patient because of a large tibial SBC that was not excised intra-operatively, causing osteolysis and metalwork loosening. During the revision surgery, the SBC was filled and packed with a platelet-rich, plasma-laden, autologous bone graft. In addition, the presence of subchondral acetabular oedema or the presence of SBCs in patients suffering from femoral–acetabular impingement at the hip joint is correlated with inferior outcomes after arthroscopic treatment compared to patients without SBCs [47]. Consequently, patients who have a combination of femoral–acetabular impingement and SBCs in MRIs may not be clinically improved after arthroscopic treatment, as there is a high probability of synchronous full-thickness cartilage lesions [47].

One of the most devastating postoperative complications after total joint replacement remains implant loosening. Despite the pre-operative identification of SBCs, especially in moderate to severe arthritis, the progress and enlargement of these cysts may cause further bone erosions and implant loosening. During the development of SBCs, symptoms may arise, and early interventions are needed to relieve symptoms and prevent major revision surgeries that create further bone loss. A study by Besse et al. [48] reported a series of 50 patients with total ankle replacement, with 18 of them (20 implants) requiring intervention due to periprosthetic osteolysis and surgical action to avoid the possibility of more severe mechanical complications. According to their conclusions, the application of an iliac crest autograft, calcium phosphate cement, and polymethylmethacrylate cement in SBCs revealed very poor outcomes, as they found a 79% functional and 92% radiological failure rate. Subsequently, the grafting of SBCs due to loosening after primary total ankle replacements did not prove an efficient technique for avoiding conversion to arthrodesis in 28% of cases [48].

Innovative surgical techniques have been used for the early treatment of SBCs in young patients to prevent knee pain and joint dysfunction and reduce the progress of osteoarthritis. Knee subchondroplasty was proposed as a treatment for an SBC in the distal medial femoral condyle of a 26-year-old woman with aplastic anaemia by Zeng et al. [49]. Under fluoroscopic guidance and using a free hand technique, a synthetic calcium phosphate-based bone substitute was introduced within the SBC. After a 2-year follow-up, the patient reported relief of symptoms, and the visual analogue scale was improved from seven preoperatively to zero postoperatively [49]. In addition, subchondroplasty has been recognised as a minimally invasive procedure with rare complications (i.e., extravasation of cement) and has a promising preventative role in halting the progress of OA.

The application of novel therapeutic interventions aiming at the treatment of SBCs and the deceleration of OA progress has been interpreted in experimental animal models as well as in humans with chronic degenerative arthritis. The retrospective study of Wallis et al. [50] reported that a single injection of corticosteroids through the fibrous lining in SBCs of the medial femoral condyle of 52 horses was a good alternative to surgical debridement, being negatively related to the presence of pre-operative osteophytes on radiographs 90 days postoperatively. The technique was based on previous findings that SBCs promoted the secretion of inflammatory mediators associated with osteoclast activation and bone absorption. According to the results of this study, arthroscopic injection with corticosteroids under arthroscopic management was linked to faster rehabilitation, earlier return to activity and lower rates of cystic enlargement [50]. Recently, the application of allogeneic chondrocyte and insulin-like growth factor-1 (IGF-1) grafting after surgical excision of SBCs of the medial femoral condyle in mature horses was proposed by Ortved et al. [51]. Based on the published literature, IGF-1 enhances the synthesis of extracellular matrix by increasing the production of the large aggregating proteoglycan (aggrecan), collagen type II as well as the link protein and hyaluronan in monolayer and 3-dimensional chondrocyte cultures. Subsequently, the application of chondrocytes and IGF-1 has proven to provide significantly higher clinical long-term outcomes based on the treatment of SBCs and prevention of OA compared to simple intralesional debridement of SBCs [51]. The remarkable effects of IGF-1 on cartilage degeneration and SBC treatment were further investigated by Zhang et al. [52] in OA rabbits. This experimental in vivo trial included the application of IGF-1-loaded collagen membranes retrieved from Achilles’ tendons to cover full-thickness articular cartilage defects [52]. According to the authors’ findings, the use of high-dose IGF-1 on collagen membranes upregulated the production of COL2A1 and enhanced the formation and survival of chondrocyte cells as well as improving the integration of neo-cartilage [52].

Tellegen et al. [53] conducted a novel and innovative experiment evaluating the effect of controlled release of celecoxib as a strategy to reduce disease progression in OA rat models. After the anterior cruciate ligament was transected and OA was developed in the knee joints of rats, unloaded microspheres carrying celecoxib were injected in the knee and micro-computed tomography, as well as histology examination, was performed post-mortem [53]. Based on the authors’ findings, celecoxib inhibited the progression of the OA phenotype by decreasing subchondral bone sclerosis, the presence of calcified loose bodies and osteophytes, as well as the formation of SBCs [53]. Celecoxib downregulates the expression of cyclo-oxygenase-2, which is responsible for the secretion of proinflammatory mediators, such as PGE2 and transcription factor NF-κB [53]. Therefore, the administration of celecoxib in OA joints not only acts as a painkiller but also reduces the inflammation of the synovium and peri-articular bone changes, such as SBC formation, representing a promising and novel agent against the progression of OA.

## 4. Conclusions

In conclusion, SBCs do not appear to significantly affect the clinical presentation in patients with OA. However, it should be stressed that their appearance as an imaging finding indicates advanced disease, which usually does not correspond to the routine means of conservative treatment. As to their significance in surgery (arthroplasty), their existence does not appear to be correlated with the overall success of the operation and the functional outcome. In addition, their existence has not so far been associated with the proper incorporation and durability of implants over time, as small to moderate cysts appear to regress after surgery and the restoration of the loads on the joint, while larger ones usually remain unchanged in size. However, immunohistological and experimental findings suggested that the early detection and management of SBCs prevented the osteochondral junction by further OA degenerative changes.

## Figures and Tables

**Figure 1 jcm-12-00815-f001:**
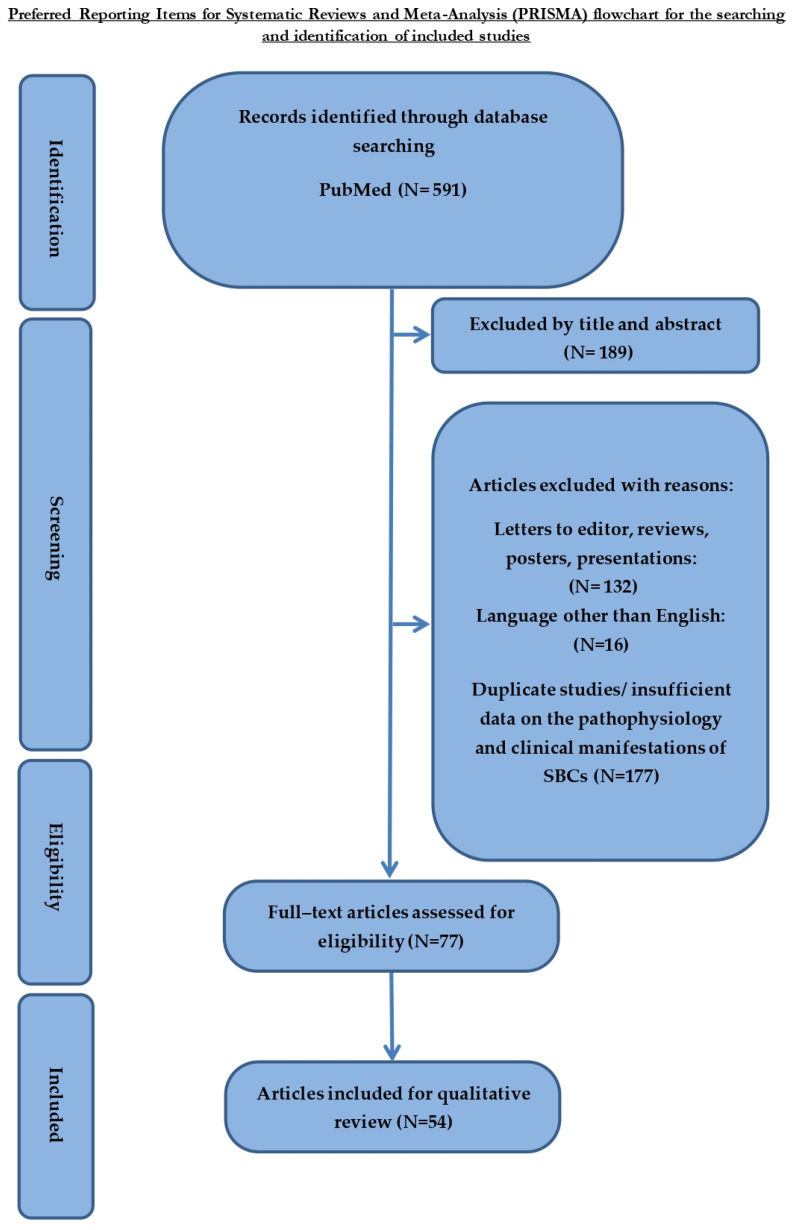
PRISMA flow chart.

**Figure 2 jcm-12-00815-f002:**
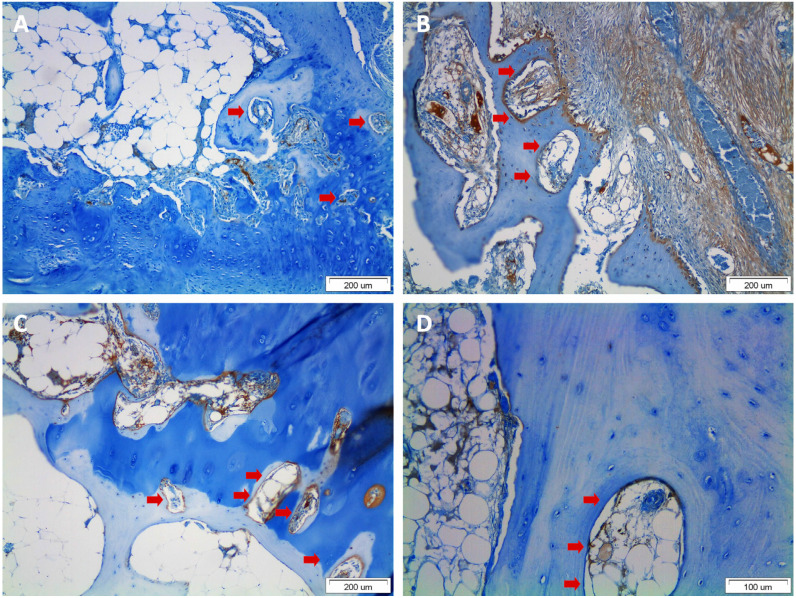
Representative immunohistological photomicrographs of subchondral bone cysts (SBCs) in sections of osteoarthritis patients. Note the thin layer of fibrous connective tissue covered by lining cells and osteoblasts. (**A**) Small-sized SBCs (red arrows) in the section of a patient with Mankin score 3 (original magnification × 10). (**B**) Large SBCs containing fibrous tissue surrounded by osteoblasts in the section of a patient with a Mankin score of 8 (original magnification × 10). (**C**) SBCs in a slice from a patient with a Mankin score of 7 (original magnification × 10). (**D**) Bone lining cells covering the cavity of a large SBC in a patient with a Mankin score of 8 (original magnification × 20).

**Figure 3 jcm-12-00815-f003:**
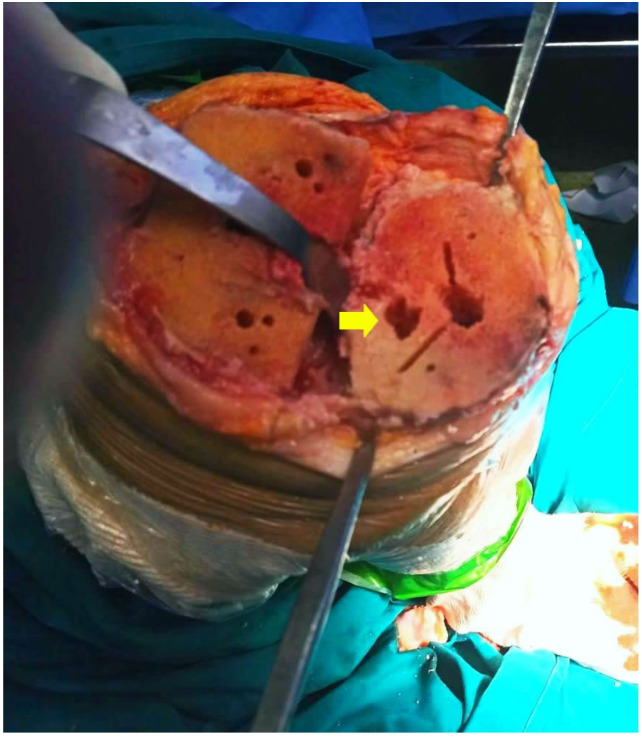
Intraoperative image during total knee arthroplasty demonstrating the gross appearance of a large subchondral bone cyst (yellow arrow). The remaining holes are the result of the femur and tibia preparation before implantation.

**Figure 4 jcm-12-00815-f004:**
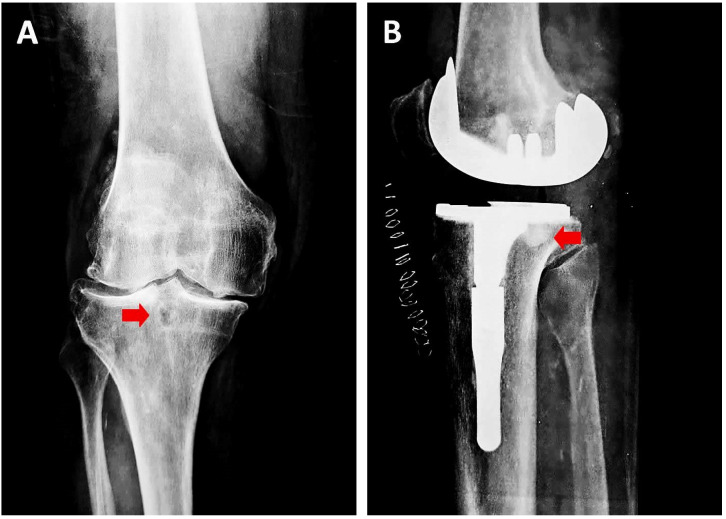
(**A**) Pre-operative anteroposterior radiograph of an osteoarthritic knee displaying the characteristic appearance of a subchondral bone cyst (red arrows). Note the sclerotic osseous margins (**B**): Postoperative lateral X-ray. After subchondral cyst excision, the gap was filled with cement.

**Table 1 jcm-12-00815-t001:** Summary of studies examining bone structural and metabolic alterations in OA in the presence of SBCs.

	Author and Year	Objective	Type of Study and Materials	Methods and Measurements	Follow-Up	Baseline	Results	Conclusions
1	Nakasone et al., 2022, [27]	The aetiology and origin of SBCs in hip OA	Observational study on femoral heads which were collected from 34 patients during THA	MCT, histological and immune-cytological analysis	N/A	N/A	SBCs in 91% of femoral heads, location: mostly close surface of the femoral head Few correlations BMI, age and sex Composition: fibrous (most common) or fatty Containing: vasculature, nerve fibres, cartilage islands, and bony spicules High-density bone, adjacent to SBCs Lower cartilage thickness on regions overlying SBCs mCT: stiffness of the primary compressive group was mildly affected by SBCs	Structural changes of SBCs: extensive perturbations in cellular activity, culminating in a multitude of osseous tissue types with heterogeneous changes in bone and cartilage morphology, which favour OA progression.
2	Hick et al., 2021, [28]	Correlation between serum levels of Coll2-1 and Coll2-1NO2 with MRI findings and clinical outcomes in knee OA.	Prospective study 121 pts with knee OA	Pts followed up for 1 year with pain, function, and MRI assessment (PRODIGE study). Type II collagen-specific biomarkers Coll2-1 and Coll2-1NO2 were directly measured in serum using immunoassays at baseline and after 3-, 6-, and 12-month follow-ups.	One-year follow-up for pain, function, and MRI assessment (PRODIGE study) Serum biomarkers: serum using immunoassays at baseline and after 3-, 6-, and 12-month follow-ups	N/A	Coll2-1 significantly correlated with periarticular cysts/bursitis, subarticular bone attrition, SBCs and articular cartilage integrity Coll2-1NO2 correlated with worse symptoms from patellofemoral, and medial femorotibial compartments, loosening bodies and osteophytes	Serum cartilage biomarkers Coll2-1 and Coll2-1NO2 are associated with several knee OA features The baseline value of Coll2-1NO2 is positively correlated with pain worsening
3	Wang et al., 2020, [29]	To discuss the mechanical function of SBCs and their relationship with Wolff’s law	Retrospective 140 symptomatic OA knees (120 pts) scheduled for HTO	Baseline MRI before HTO SBCs and BMLs count, Hip–knee–ankle axis pre-op and evolution after HTO	5 years	SBCs were detected in 72 knees, with 70 (97%) showing a BML surrounding the SBC	Average vol 9.6 ± 4.1 mm^3^ with adjacent cartilage present with no full-thickness defects Mean HKA axis was 7.3 ± 3 degrees of varus After 5 y regression of SBCs in 50 knees → revision for TKA 2/50 (4%), no regression 22 knees → revision for TKA 18/22 (82%) Mean SBC vol for regression group after 5 y → 6.3 ± 2.8 mm^3^	Regression of SBCs may be related to the restoration of an appropriate load
4	Lenz et al., 2019, [30]	To investigate if acetabular SBCs consolidate spontaneously after THA	Retrospective Primary THA for 54 hip joints in 52 pts	AP hip radiograph preop, postop and at the latest follow-up Number of SBCs, size and progression	6.3 years (5–9)	88 cysts with a mean size of 9.3 ± 10 mm^2^ (range 0.9–57 mm^2^)	71 SBCs (80.7%) in 38 hips disappeared in follow-up 17 SBCs (19.3%) in 16 hips are still visible (mostly in Charnley zone I) 15/17 SBCs decreased from 19.8 ± 18 mm^2^ preop to 13.3 ± 13.3 mm^2^ postop 2/17 SBCs progressed from 8.4 ± 2 preop to 15.8 ± 10.9 mm^2^ postop	Post THR, most neglected SBCs decrease in size. Larger cysts may persist without impact on surgical outcomes. No radiological signs of loosening were observed when acetabular SBCs are neglected during primary THR.
5	Takada et al., 2017, [31]	To clarify morphological changes of acetabular SBCs after THA for OA secondary to dysplasia	Retrospective 261 primary uncemented THA in 208 pts	Baseline CT preop Number of SBCs and evolution	7–10 years (mean 8.4)	Preop CT → SBCs detected in 128 cases (49%)	Mean cross-sectional area of SBCs 3 m postop was 159.87 ± 130.05 mm^2^ No new SBC formation in 7–10-year follow-up Mean cross-sectional area of SBCs 7–10 y postop was 110.18 ± 130.05 mm^2^ (mean regression of 70.0 ± 51.6 mm^2^)	Acetabular SBCs do not expand after THA. The longitudinal morphological change of acetabular SBCs does not influence long-term implant fixation in THA.
6	Mechlenburg et al., 2012, [32]	To investigate whether the volume of acetabular and femoral head SBCs would change after periacetabular osteotomy	Prospective 26 pts scheduled for PAO	MRI preop Number of SBCs and total cyst volume. Pts also filled HOOS 4 years after PAO	1 and 2.5 as well as 10 years	22 SBCs in 12 pts (21 acetabular)	Mean total acetabular SBC volume per pt = 3.44 cm^3^ SD = 6.71 23 SBCs in 15 pts 1-year postop and 18 SBCs in 15 pts 2,5 years postop Mean total acetabular SBC volume per pt decreased → to 1.96 cm^3^ SD 3.97 1 y postop → to 0.96 cm^3^ SD 1.70 2, 5 y postop	No. of pts having SBCs did not change notably but the mean total cyst volume /pt decreased significantly
7	Tanamas et al., 2010, [19]	To determine the correlation between cartilage loss and risk of joint replacement with SBCs and BMLs	Retrospective Symptomatic knee in 132 pts	Baseline MRI Tibial cartilage volume, SBCs, BMLs, Risk for TKR over 4 years	2 years	BSCs in 47.7% of subjects 98.1 of which had also BMLs	Over 2 y, 23.9% progress, 13% new SBC, 11.4% regress Baseline presence of BSC associated with lower tibial cartilage vol compared to those having only BML or neither	Annual cartilage loss is greatest in those with BSC compared to BML or neither As severity increased from no BMLs or SBCs → only BMLs → SBCs the risk of TKR increased
8	Kelly et al., 2007, [33]	To quantify the radiographic changes of OA acetabular SBCs after uncemented THA	Retrospective 130 primary uncemented THA	Baseline postop radiographs SBCs count and 2-dimensional size	10 years	41 SBCs identified postop with a mean size of 1 ± 0.9 cm^2^ (range 0.2–4.5)	4 SBCs (10%) expanded after 10 y with an average increase in the size of 5.1 ± 8.6 cm^2^ (713%) 27 SBCs (66%) regressed. 23 (56%) disappeared at a mean of 5.2 ± 2.7 years (2/23 had been grafted) 10 SBCs (24%) quasi-static	No difference among the 3 SBCs groups in initial size, pt age, pt weight, different cup types, pt sex, and bone graft The only variable correlated was SBC location → zone II more likely to progress (75% of SBCs that progressed were in zone II) Recommendation to place the cup to seal SBC + grafting at the time of primary THA to prevent progression and osteolysis

## Data Availability

Not applicable.

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
