# Peer review of "Subchondral Bone Cyst Development in Osteoarthritis: From Pathophysiology to Bone Microarchitecture Changes and Clinical Implementations"

_jcm, 2023, doi:10.3390/jcm12030815_

Round 1

Reviewer 1 Report

Dear authors, i read the manuscript with great interest as it deals with an important issue in daily orthopaedic practice. The manuscript is well written and results are well supported.

Author Response

We would like to thank the reviewer for that kind comment and we are very grateful that you enjoyed our manuscript.

Reviewer 2 Report

I want to congratulate the authors of this review manuscript. It is an exciting and refreshing review with a lot of potential. Here are my comments:

Abstract

This section is less developed than the others. Especially, English grammar, cohesion, and coherence are below the rest of the manuscript. I would suggest it be rewritten from scratch.

Introduction

It is an interesting introduction, nonetheless, coherence and cohesion are missing. Several statements are written quite confusingly. Specifically, the introduction needs to be devoid of opinions, and suggestions. The state of the art needs to be perfectly cited. Here are two examples:

·       MRIs are considered the examination of choice for the imaging of articular cartilage and bone marrow lesions.

·        According to this observation, areas of bone oedema can be characterized as pre-cystic changes.

Is this an opinion of the group? A consensus in the field but not yet proven? Is it ton debate? Or it has been proven enough to appear in the medical guidelines. Please cite accordingly.

Please revise the whole section.

M&M

The manuscript needs to elaborate more on the inclusion and exclusion criteria. These include the proper “Inclusion & Exclusion criteria” section and the following paragraph “In addition, the screening of articles and abstracts was performed by the two senior authors (A.K., and A.C.H.), independently. All identified studies were evaluated by the authors (I.D.I., D.N., I.L., and D.M.) and data were extracted using a predetermined form”

Publication standards for systematic reviews have risen to facilitate data transparency and comparison, and avoid work duplications. In this sense, a pipeline on how many manuscripts were initially found and how many were discarded per each different criterion is mandatory. Most authors build a tree-branch diagram. Most also include the PMIDS or DOIs of the included and excluded works. Moreover, I came across that the authors explicitly mentioned that they included “reviews or clinical studies” as well as “In vitro and in vivo experimental studies” which are typically referred to as preclinical studies. This sort of definition would automatically discard “in silico” studies such as those using computational chemistry, neural networks, and so on. Furthermore, it is necessary to explicitly mention whether case reports were included or excluded.

Results

It is a brilliant section. I want to congratulate the authors for this section. It is very informative plus the subsections are clear and logical. Nonetheless, there is a single minor concern. Nearly all statements are solely supported by a single manuscript or a single citation. As an example:

“Therefore enhancing the healing of damaged and the formation of new cartilage [52]”

If the exclusion criteria only left this manuscript to back up this statement, it is fine once the Inclusion & Exclusion criteria section is extended. Nonetheless, there are solid proof statements supported with one citation and a debatable statement (like the previous example that is clinically on debate). If the bulk of citations do not reflect how accepted it is as a statement, language should reflect that.

Conclusions

No comments

Author Response

Reviewer #2: I want to congratulate the authors of this review manuscript. It is an exciting and refreshing review with a lot of potential. Here are my comments:

Abstract

This section is less developed than the others. Especially, English grammar, cohesion, and coherence are below the rest of the manuscript. I would suggest it be rewritten from scratch.

Answer: Thank you for pointing this out. We have, accordingly revised and improved the quality of the abstract from scratch. In addition, the manuscript has been checked by a native English-speaker.

Introduction

It is an interesting introduction, nonetheless, coherence and cohesion are missing. Several statements are written quite confusingly. Specifically, the introduction needs to be devoid of opinions, and suggestions. The state of the art needs to be perfectly cited. Here are two examples:

  • MRIs are considered the examination of choice for the imaging of articular cartilage and bone marrow lesions.
  • According to this observation, areas of bone oedema can be characterized as pre-cystic changes.

Is this an opinion of the group? A consensus in the field but not yet proven? Is it ton debate? Or it has been proven enough to appear in the medical guidelines. Please cite accordingly.

Please revise the whole section.

Answer: We would like to thank the reviewer for this suggestion. These statements have been revised and the whole manuscript has been revised by a certified translator and native British English-speaker

M&M

The manuscript needs to elaborate more on the inclusion and exclusion criteria. These include the proper “Inclusion & Exclusion criteria” section and the following paragraph “In addition, the screening of articles and abstracts was performed by the two senior authors (A.K., and A.C.H.), independently. All identified studies were evaluated by the authors (I.D.I., D.N., I.L., and D.M.) and data were extracted using a predetermined form”

Publication standards for systematic reviews have risen to facilitate data transparency and comparison, and avoid work duplications. In this sense, a pipeline on how many manuscripts were initially found and how many were discarded per each different criterion is mandatory. Most authors build a tree-branch diagram. Most also include the PMIDS or DOIs of the included and excluded works. Moreover, I came across that the authors explicitly mentioned that they included “reviews or clinical studies” as well as “In vitro and in vivo experimental studies” which are typically referred to as preclinical studies. This sort of definition would automatically discard “in silico” studies such as those using computational chemistry, neural networks, and so on. Furthermore, it is necessary to explicitly mention whether case reports were included or excluded.

Answer: We agree to the reviewer’s opinion on the materials and methods section. This section has been revised based on the reviewer’s guidance and a flow chart is added to clarify the number of included and excluded articles.

Results

It is a brilliant section. I want to congratulate the authors for this section. It is very informative plus the subsections are clear and logical. Nonetheless, there is a single minor concern. Nearly all statements are solely supported by a single manuscript or a single citation. As an example:

“Therefore enhancing the healing of damaged and the formation of new cartilage [52]”

If the exclusion criteria only left this manuscript to back up this statement, it is fine once the Inclusion & Exclusion criteria section is extended. Nonetheless, there are solid proof statements supported with one citation and a debatable statement (like the previous example that is clinically on debate). If the bulk of citations do not reflect how accepted it is as a statement, language should reflect that.

 Answer: We would like to thank the reviewer for this suggestion. The abovementioned sentence (“Therefore…new cartilage”) has been removed from the article as is not clarified and supported in the article by Ortved et al (reference 52). 

Conclusions

No comments